# π-Extended Boron Difluoride [N∩NBF₂] Complex, Crystal Structure, Liquid NMR, Spectral, XRD/HSA Interactions: A DFT and TD-DFT Study

**Abdulrahman A. Alsimaree [1,\*], Nawaf. I. Alsenani [2], Omar Mutlaq Alatawi [3,4], Abeer A. AlObaid [5], Julian Gary Knight [3] , Mouslim Messali [6], Abdelkader Zarrouk [7] and Ismail Warad [8,\*]**

1 Department of Basic Science (Chemistry), College of Science and Humanities, Shaqra University, Afif, P.O. Box 33, Shaqra 11961, Saudi Arabia

2 Department of Chemistry, Faculty of science, University of Albaha, Alagig 65779-7738, Saudi Arabia; N.alsenani@bu.edu.sa

3 Department of Chemistry, School of Natural and Environmental Sciences, Newcastle University, Newcastle Upon Tyne NE1 7RU, UK; omalatawi@gmail.com (O.M.A.); julian.knight@newcastle.ac.uk (J.G.K.)

4 Department of Chemistry, Faculty of science, University of Tabuk, Tabuk 47512, Saudi Arabia

5 Department of Chemistry, College of Science, King Saud University, P.O. Box 2455, Riyadh 11451, Saudi Arabia; aaalobaid@ksu.edu.sa

6 Laboratory of Applied and Environmental Chemistry (LCAE), Mohammed First University, Oujda 60000, Morocco; mouslim.messali@gmail.com

7 Laboratory of Materials, Nanotechnology, and Environment, Faculty of Sciences, Mohammed V University in Rabat, Agdal-Rabat P.O. Box. 1014, Morocco; azarrouk@gmail.com

8 Department of Chemistry and Earth Sciences, Qatar University, Doha P.O. Box 2713, Qatar

\* Correspondence: alsimaree@su.edu.sa (A.A.A.); ismail.warad@qu.edu.qa (I.W.)

**Abstract:** The novel tetrahedral 10-(4-carboxyphenyl)-2,8-diethyl-5,5-difluoro-1,3,7,9-tetramethyl-5H-di-pyrrolo[1,2-c:2',1'-f][1,3,2]diazaborinin-4-ium-5-uide [N∩NBF₂] BODIPY complex was prepared in a very good yield and via one-pot synthesis. The desired [N∩NBF₂] has been used as a model complex for XRD/HSA interactions and DFT/B3LYP/6-311G(d,p) computations. The tetrahedral geometry around the boron center was demonstrated by DFT optimization and XRD-crystallography. The $^1$H, $^{11}$B, and $^{19}$F-NMR spectra were used also to support the high symmetrical BODIPY via π-extended phenomena. Moreover, the values of the DFT-calculated structural bond lengths/angles and DFT-IR were matched to the corresponding experimental XRD and IR parameters, respectively. The crystal lattice interactions were correlated to Hirshfeld surface analysis (HSA) calculations. Calculations of the Mulliken Atomic Charge (MAC), Natural Population Analysis (NPA), Global reactivity descriptors (GRD), and Molecular Electrostatic Potential (MEP) quantum parameters were performed to support the XRD/HSA interactions result. Analysis of the predicted Density of States (DOS), molecular orbital, and time-dependent density functional theory (TD-DFT) calculations have been combined to explain the experimental UV-vis spectra and electron transfer behavior in [N∩NBF₂] complex using MeOH and other four solvents.

**Keywords:** XRD; B-complex; HSA; NMR; TD-DFT/UV-vis.

## 1. Introduction

Complexes of boron with dipyrromethene ligands (BODIPY) as a class of organic complexes are currently of high interest due to their uses in several dye applications such as fluorescent switches, biomolecule markers, organic solar cells, chemosensors, laser dyes, photodynamic therapy, and fluorescence surface labeling [1–5]. Moreover, there has been a rapid growth of new preparation strategies for the functionalization of B-complexes to enable binding to a biological target in order to change its optical properties; these strategies include cross-coupling reaction, halogenation, and nucleophilic aromatic substitution on the dipyrromethene part [6–8]. The big challenges in the chemistry of BODIPY are in

developing compounds with enhanced emission and absorption profiles and the discovery of dyes with new properties [9–11].

The ongoing accommodation is explicit from the considerable number of articles being reported about the synthesis and characterization of BODIPY boron complexes as well as their varied applications such as biological labels, tunable laser dyes, probes, photoactive, photovoltaic devices, fluorescent nanocars, light-emitting devices, photoactivatable compounds, energy transfer, triplet photosensitizers, PDT, photocatalytic reactions, triplet–triplet annihilation up-conversion, and photo-induced production [12–26]. Moreover, various BODIPY dyes are commercially available as probes for bio-imaging, biological labels, and laser dyes [27,28]. Recently, the same complex and its analogs were prepared by Shipalova et al.; the author's studies focused on the pH and polarity fluorescent molecular sensorics involving the BODIPY ligand. Moreover, none of the complexes structures were solved by XRD crystal nor was the $\pi$-extended boron difluoride phenomena studied [28].

There are a considerable number of papers concerning the BODIPY difluoride complexes, but few of them have been supported by X-raying a single crystal; therefore, herein, we have synthesized new [N∩NBF$_2$] and characterized the structure by XRD. The solution-phase NMR and UV-Vis. spectra helped in supporting the $\pi$-extended boron difluoride phenomena in the [N∩NBF$_2$] complex. The interactions in the structure have been confirmed by XRD/HSA analysis; moreover, the DFT/XRD structural parameters were successfully matched. The results of TD-DFT calculations, including methanol solvation, agreed well with the experimental UV-Vis behavior of the [N∩NBF$_2$] complex.

## 2. Experimental

### 2.1. Computational

The HSA was carried out through the CIF file data and using the Crystal Explorer 3.1 program (version 17, University of Western Australia, Perth, Australia) [29]. All the DFT calculations were performed in gaseous phase at the DFT/B3LYP/6-311G(d,p) level of theory *via* Gaussian09 software (version 09, Gaussian Inc., Wallingford, CT, USA) [30].

### 2.2. Materials and Synthesis

Commercially available materials and solvents used in this study were purchased from commercial suppliers (Sigma Aldrich, Fluorochem and Alfa Aesar). The desired $\pi$-extended boron difluoride [N∩NBF$_2$] complex has been prepared according to the published procedure [30]. The NMR was performed on a JEOL ECS 400 MHz Bruker Advance 300 (Bruker GmbH, Berlin, Germany) instrument using CDCl$_3$ as solvent at RT. First, 10 mg of the complex powder was suspended in 3 ml of CDCl$_3$, the clean solution was decanted to 3mm Wilmad NMR tube and filled up to $\approx$3 cm length to be used for the H, B, and F NMR. Since NMR showed that the elements H, B, and F have high natural abundance <99%; therefore, classical NMR was performed with 90° pulse and 8 to 32 scans and TMS reference for $^1$H-NMR, CFCl$_3$ reference for $^{19}$F-NMR and BF$_3$.OEt$_2$ for $^{11}$B-NMR were used with UV-Vis on a TU-1901 double-beam UV-Vis spectrophotometer (PerkinElmer Inc., Waltham, MA, USA).

### 2.3. XRD-Structure

The single crystal X-ray data of the complex were collected on a Bruker D8 Quest diffractometer (Bruker GmbH, Berlin, Germany) (Mo-K$\alpha$ radiation $\lambda$ = 0.71073 Å). The structure was solved by direct methods using SHELXS and SHELXTL packages (Uni. Gottingen, Gottingen, Germany), which were refined using full-matrix least squares procedures on $F^2$ via the program SHELXL [31,32]. All hydrogen atoms were included at calculated positions using a-riding model with C–H distances of 0.93 Å for sp$^2$ carbons and 0.96–0.97 Å for sp$^3$ carbons. The isotropic displacement parameters were U$_{iso}$(H) = 1.2U$_{eq}$(C) for methylene groups and sp$^2$ carbons and U$_{iso}$(H) = 1.5U$_{eq}$(C) for methyl groups. The crystal data and refinement details are given in Table 1.

**Table 1.** [N∩NBF$_2$] crystal refinement data.

| Empirical Formula | C$_{24}$H$_{27}$BF$_2$N$_2$O$_2$ |
| --- | --- |
| Formula weight | 424.28 |
| Temperature/K | 150.0(2) |
| Crystal system | Triclinic |
| Space group | P-1 |
| a/Å | 7.6047(3) |
| b/Å | 7.7074(3) |
| c/Å | 21.7634(10) |
| α/° | 89.751(4) |
| β/° | 84.955(3) |
| γ/° | 83.375(3) |
| Volume/Å$^3$ | 1262.16(9) |
| Z, Z′ | 2, 1 |
| ρ$_{calc}$ g/cm$^3$ | 1.116 |
| μ/mm$^{-1}$ | 0.660 |
| F(000) | 448.0 |
| Crystal size/mm$^3$ | 0.27 × 0.16 × 0.04 |
| Radiation | CuKα (λ = 1.54184) |
| 2Θ range for data collection/° | 11.558 to 133.652 |
| Index ranges | −9 ≤ h ≤ 8, −9 ≤ k ≤ 9, −25 ≤ l ≤ 25 |
| Reflections collected | 17660 |
| Independent reflections | 4444 [R$_{int}$ = 0.0465, R$_{sigma}$ = 0.0367] |
| Data/restraints/parameters | 4444/0/292 |
| Goodness-of-fit on F$^2$ | 1.025 |
| Final R indexes [I >= 2σ (I)] | R$_1$ = 0.0452, wR$_2$ = 0.1163 |
| Final R indexes [all data] | R$_1$ = 0.0592, wR$_2$ = 0.1257 |
| Largest diff. peak/hole/e Å$^{-3}$ | 0.23/−0.23 |

## 3. Results and Discussion

### 3.1. Preparation and NMR

The desired [N∩NBF$_2$] complex was synthesized according to the reported method [30] as shown in Scheme 1. The negative charge of the dipyrromethene ligand can be fully delocalized across the extended π-system, and this is expected to result in a symmetrical electron distribution between N1 and N2 which are bridged by the boron center, as shown in Scheme 1.

**Scheme 1.** Synthesis of the [N∩NBF$_2$] complex.

The experimental solution-phase $^1$H, $^{11}$B and $^{19}$F-{$^1$H}-NMR spectra in CDCl$_3$ support the effective conjugation of organic π-bonds in the BODIPY in system to form a highly symmetrical hybrid structure (Scheme 1 and Figure 1). The δ$_v$-symmetry reduces the number of protons environment to 6 instead of 12; all the protons groups chemical shifts are identified directly and labeled in the spectrum in Figure 1a. Moreover, as expected, the $^{11}$B-{$^1$H}-NMR spectrum displays a triplet (128 Mhz, δ = 1.0 ppm), as seen in Figure 1b due to

coupling to two equivalent fluorine atoms, and the $^{19}$F-{$^1$H}-NMR shows at 1:1:1:1 quartet (376.5 MHz, δ = −146.3 ppm), resulting from two equivalent fluorine atoms due the π-extended behavior coupled to the boron [33,34]. NMRDB [35] and GIAO-DFT/B3LYP/6-311G(d,p) NMR [30] were performed using same references and CDCl$_3$ solvent, and the results are inserted into Figure 1 together with their experimental relatives. The difference in the theoretical calculations is not surprising, because in theory, NMR cannot determine the π-extended phenomena; therefore, the BODIPY in the [N∩NBF$_2$] complex becomes an asymmetrical ligand and has no-C2-plane of symmetry that can be seen from the proton chemical shifts in Figure 1a,b. The NMRDB $^1$H-NMR chemical shifts and splitting together with the Exp. NMR are in good agreement. The only difference is that the NMRDB-$^1$H-NMR perceived protons of 3 and of 4 groups as being identical and having the same chemical shift. Meanwhile, in $^{11}$B-NMR (Figure 1d) and $^{19}$F (Figure 1f), the DFT neglected the effect of the two nuclei on each other; therefore, no splitting was detected, resulting in a simple system: singlet for B atoms and doublet for F atoms.

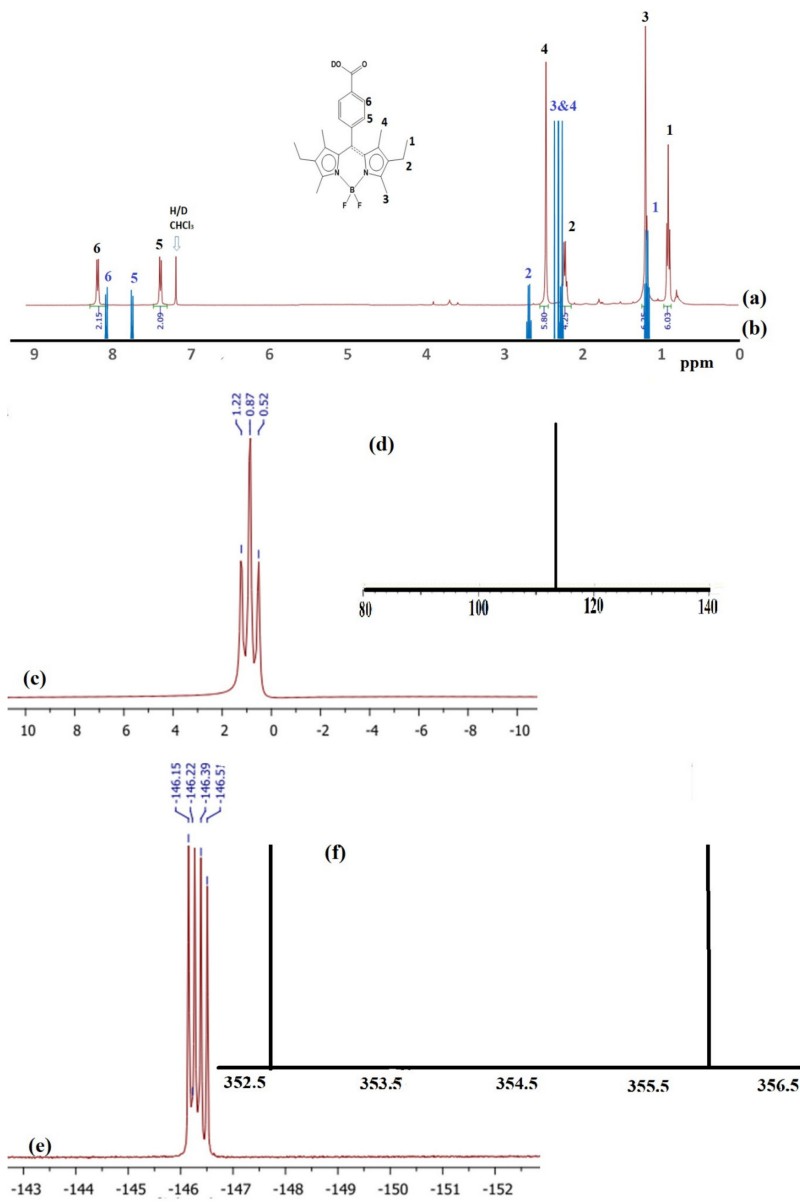

**Figure 1.** (**a**) Exp. $^1$H and (**b**) NMRDB–$^1$H-NMR, (**c**) $^{11}$B-{$^1$H } and (**d**) GIAO-NMR, (**e**) $^{19}$F-{$^1$H}- and (**f**) GIAO-NMR in CDCl$_3$.

### 3.2. XRD and DFT

The formation of the desired boron complex was confirmed via XRD analysis, as shown below. Analysis and comparison between the experimentally determined XRD and the computed DFT/B3LYP/6-311G(d,p)-optimized structures and their structural parameters is discussed below.

The B-complex was crystalized in a Triclinic/P-1 system; the molecular structures along with angles and bond lengths are illustrated in Figure 2a and Table 2. Figure 2a and Table 2 indicate the expected tetrahedral geometry of the boron center with N,N-chelation and 2F ligands [N-B-N angle (107.2°) F-B-F angle (109.5°)]. Formally, the boron center is coordinated to one of the N atoms via an ionic bond and to the other via a coordination co-valent bond to form a slightly strained-six membered ring that is perpendicular to the plane of the F-B-F group. The 2B-N (1.543 Å), B-F1 (1.397 Å), and B-F2 (1.389 Å) bond lengths are consistent with the reported data for similar tetra-coordinate boron systems [32–38]. The DFT theoretical calculations demonstrated a high degree of congruence, as seen in Figure 2b. Quantification of the level of agreement between the theoretical and experimental bonds lengths and angles is shown in Table 2.

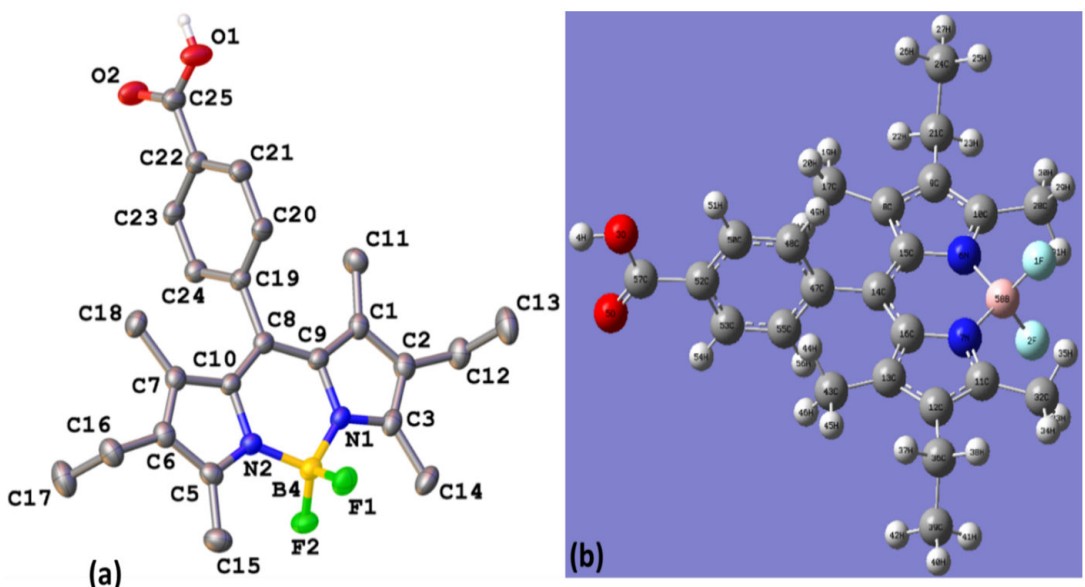

**Figure 2.** Tetrahedral of B-complex structures: (**a**) ORTEP and (**b**) DFT-optimized.

**Table 2.** DFT/XRD bond lengths (Å), angles (°), and dihedral angles (°).

| No. | Bond | | XRD | DFT | No. | Bond | | XRD | DFT |
|---|---|---|---|---|---|---|---|---|---|
| 1 | F1 | B4 | 1.397(2) | 1.4018 | 18 | C5 | C6 | 1.414(3) | 1.4179 |
| 2 | F2 | B4 | 1.389(2) | 1.4017 | 19 | C5 | C15 | 1.493(3) | 1.4927 |
| 3 | O1 | H1 | 0.830(8) | 0.9681 | 20 | C6 | C7 | 1.387(2) | 1.3961 |
| 4 | O1 | C25 | 1.264(3) | 1.3555 | 21 | C6 | C16 | 1.497(3) | 1.5042 |
| 5 | O2 | C25 | 1.263(2) | 1.2074 | 22 | C7 | C10 | 1.434(2) | 1.4341 |
| 6 | N1 | C3 | 1.353(3) | 1.3441 | 23 | C7 | C18 | 1.496(3) | 1.5002 |
| 7 | N1 | C9 | 1.399(2) | 1.3963 | 24 | C8 | C9 | 1.407(2) | 1.4033 |
| 8 | N1 | B4 | 1.543(3) | 1.5531 | 25 | C8 | C10 | 1.389(2) | 1.4033 |
| 9 | N2 | C5 | 1.345(3) | 1.3442 | 26 | C8 | C19 | 1.493(2) | 1.4942 |
| 10 | N2 | C10 | 1.400(2) | 1.3963 | 27 | C12 | C13 | 1.519(4) | 1.5393 |
| 11 | N2 | B4 | 1.543(3) | 1.553 | 28 | C16 | C17 | 1.519(3) | 1.5393 |
| 12 | C1 | C2 | 1.393(3) | 1.3962 | 29 | C19 | C20 | 1.385(2) | 1.3985 |
| 13 | C1 | C9 | 1.419(2) | 1.4341 | 30 | C19 | C24 | 1.397(2) | 1.3994 |
| 14 | C1 | C11 | 1.501(3) | 1.5002 | 31 | C20 | C21 | 1.382(2) | 1.3904 |
| 15 | C2 | C3 | 1.403(3) | 1.4179 | 32 | C21 | C22 | 1.395(2) | 1.3985 |
| 16 | C2 | C12 | 1.505(3) | 1.5042 | 33 | C22 | C23 | 1.392(3) | 1.3982 |
| 17 | C3 | C14 | 1.494(3) | 1.4927 | 34 | C22 | C25 | 1.483(2) | 1.4877 |

**Table 2.** *Cont.*

| No. | Angles | | | XRD | DFT | No. | Angles | | | XRD | DFT |
|---|---|---|---|---|---|---|---|---|---|---|---|
| 1 | H1 | O1 | C25 | 117(6) | 106.26 | 28 | C10 | C8 | C19 | 119.0(1) | 119.27 |
| 2 | C3 | N1 | C9 | 108.1(1) | 108.89 | 29 | N1 | C9 | C1 | 107.9(1) | 107.55 |
| 3 | C3 | N1 | B4 | 126.3(2) | 125.29 | 30 | N1 | C9 | C8 | 119.6(1) | 120.13 |
| 4 | C9 | N1 | B4 | 125.6(1) | 125.82 | 31 | C1 | C9 | C8 | 132.5(2) | 132.32 |
| 5 | C5 | N2 | C10 | 108.3(1) | 108.89 | 32 | N2 | C10 | C7 | 107.5(1) | 107.55 |
| 6 | C5 | N2 | B4 | 126.1(2) | 125.29 | 33 | N2 | C10 | C8 | 120.1(1) | 120.13 |
| 7 | C10 | N2 | B4 | 125.4(1) | 125.82 | 34 | C7 | C10 | C8 | 132.4(2) | 132.32 |
| 8 | C2 | C1 | C9 | 106.9(1) | 106.73 | 35 | C2 | C12 | C13 | 112.5(2) | 113.71 |
| 9 | C2 | C1 | C11 | 124.5(2) | 124.86 | 36 | C6 | C16 | C17 | 112.2(2) | 113.7 |
| 10 | C9 | C1 | C11 | 128.7(2) | 128.41 | 37 | C8 | C19 | C20 | 120.7(1) | 120.37 |
| 11 | C1 | C2 | C3 | 107.5(2) | 107.26 | 38 | C8 | C19 | C24 | 119.6(1) | 120.42 |
| 12 | C1 | C2 | C12 | 127.6(2) | 127.48 | 39 | C20 | C19 | C24 | 119.6(2) | 119.21 |
| 13 | C3 | C2 | C12 | 124.9(2) | 125.25 | 40 | C19 | C20 | C21 | 120.5(2) | 120.55 |
| 14 | N1 | C3 | C2 | 109.6(2) | 109.57 | 41 | C20 | C21 | C22 | 119.9(2) | 120.01 |
| 15 | N1 | C3 | C14 | 122.2(2) | 121.92 | 42 | C21 | C22 | C23 | 119.9(2) | 119.61 |
| 16 | C2 | C3 | C14 | 128.1(2) | 128.51 | 43 | C21 | C22 | C25 | 119.9(2) | 122.33 |
| 17 | N2 | C5 | C6 | 109.9(2) | 109.57 | 44 | C23 | C22 | C25 | 120.2(2) | 118.06 |
| 18 | N2 | C5 | C15 | 122.7(2) | 121.92 | 45 | C22 | C23 | C24 | 119.8(2) | 120.21 |
| 19 | C6 | C5 | C15 | 127.4(2) | 128.51 | 46 | C19 | C24 | C23 | 120.2(2) | 120.41 |
| 20 | C5 | C6 | C7 | 107.3(1) | 107.26 | 47 | O1 | C25 | O2 | 123.3(2) | 122.26 |
| 21 | C5 | C6 | C16 | 125.2(2) | 125.25 | 48 | O1 | C25 | C22 | 118.1(2) | 112.86 |
| 22 | C7 | C6 | C16 | 127.4(2) | 127.48 | 49 | O2 | C25 | C22 | 118.6(2) | 124.88 |
| 23 | C6 | C7 | C10 | 107.0(1) | 106.73 | 50 | F1 | B4 | F2 | 109.5(2) | 109.76 |
| 24 | C6 | C7 | C18 | 124.4(2) | 124.86 | 51 | F1 | B4 | N1 | 109.4(2) | 110.01 |
| 25 | C10 | C7 | C18 | 128.6(2) | 128.41 | 52 | F1 | B4 | N2 | 109.5(2) | 110.19 |
| 26 | C9 | C8 | C10 | 121.9(1) | 121.48 | 53 | F2 | B4 | N1 | 110.6(2) | 110.19 |
| 27 | C9 | C8 | C19 | 119.1(1) | 119.25 | 54 | F2 | B4 | N2 | 110.6(2) | 110.02 |

| No. | Dihedral angles | | | | XRD | DFT | No. | Dihedral angles | | | | XRD | DFT |
|---|---|---|---|---|---|---|---|---|---|---|---|---|---|
| 1 | C9 | N1 | C3 | C2 | 0.5 | −0.22 | 32 | C11 | C1 | C2 | C12 | 1.5 | 0.98 |
| 2 | C9 | N1 | C3 | C14 | 178.6 | 179.71 | 33 | C2 | C1 | C9 | N1 | 0.4 | 0.07 |
| 3 | B4 | N1 | C3 | C2 | 179.1 | 179.88 | 34 | C2 | C1 | C9 | C8 | −178 | −179.78 |
| 4 | B4 | N1 | C3 | C14 | 1.8 | −0.2 | 35 | C11 | C1 | C9 | N1 | 179.4 | 179.99 |
| 5 | C3 | N1 | C9 | C1 | −0.6 | 0.09 | 36 | C11 | C1 | C9 | C8 | 2.1 | 0.13 |
| 6 | C3 | N1 | C9 | C8 | 178.1 | 179.97 | 37 | C1 | C2 | C3 | N1 | −0.3 | 0.26 |
| 7 | B4 | N1 | C9 | C1 | 179.1 | 179.99 | 38 | C1 | C2 | C3 | C14 | 178.8 | 179.65 |
| 8 | B4 | N1 | C9 | C8 | −2.3 | −0.13 | 39 | C12 | C2 | C3 | N1 | 178 | 179.19 |
| 9 | C3 | N1 | B4 | F1 | 65.7 | 60.63 | 40 | C12 | C2 | C3 | C14 | −3 | −0.72 |
| 10 | C3 | N1 | B4 | F2 | −54.9 | −60.52 | 41 | C1 | C2 | C12 | C13 | 86.7 | 89.5 |
| 11 | C3 | N1 | B4 | N2 | −175.5 | −179.89 | 42 | C3 | C2 | C12 | C13 | −91.2 | −89.21 |
| 12 | C9 | N1 | B4 | F1 | −113.8 | −119.26 | 43 | N2 | C5 | C6 | C7 | 1.3 | 0.22 |
| 13 | C9 | N1 | B4 | F2 | 125.5 | 119.59 | 44 | N2 | C5 | C6 | C16 | 178.6 | 179.12 |
| 14 | C9 | N1 | B4 | N2 | 4.9 | 0.22 | 45 | C15 | C5 | C6 | C7 | −177 | −179.69 |
| 15 | C10 | N2 | C5 | C6 | −1.4 | −0.18 | 46 | C5 | C6 | C7 | C18 | 177 | 179.85 |
| 16 | C10 | N2 | C5 | C15 | 177 | 179.73 | 47 | C16 | C6 | C7 | C10 | 177 | 179.03 |
| 17 | B4 | N2 | C5 | C6 | 174.3 | 179.81 | 48 | C6 | C7 | C10 | C8 | −177 | −179.9 |
| 18 | B4 | N2 | C5 | C15 | −7.3 | −0.27 | 49 | C18 | C7 | C10 | N2 | −177 | −179.96 |
| 19 | C5 | N2 | C10 | C7 | 1 | 0.08 | 50 | C10 | C8 | C9 | C1 | 177 | 179.8 |
| 20 | C5 | N2 | C10 | C8 | 179.4 | 179.96 | 51 | C19 | C8 | C9 | N1 | 177 | 179.95 |
| 21 | B4 | N2 | C10 | C7 | −174.8 | −179.92 | 52 | C9 | C8 | C10 | C7 | −177 | −179.96 |
| 22 | B4 | N2 | C10 | C8 | 3.6 | 0.04 | 53 | C19 | C8 | C10 | N2 | −177 | −179.9 |
| 23 | C5 | N2 | B4 | F1 | −61.9 | −60.81 | 54 | C9 | C8 | C19 | C20 | −177 | −88.83 |
| 24 | C5 | N2 | B4 | F2 | 58.8 | 60.35 | 55 | C10 | C8 | C19 | C20 | 177 | 91.17 |
| 25 | C5 | N2 | B4 | N1 | 179.4 | 179.82 | 56 | C10 | C8 | C19 | C24 | −177 | −88.84 |
| 26 | C10 | N2 | B4 | F1 | 113.1 | 119.19 | 57 | C8 | C19 | C20 | C21 | −177 | −179.97 |
| 27 | C10 | N2 | B4 | F2 | −126.2 | −119.66 | 58 | C8 | C19 | C24 | C23 | −177 | −179.93 |
| 28 | C10 | N2 | B4 | N1 | −5.5 | −0.18 | 59 | C20 | C21 | C22 | C25 | −177 | −179.94 |
| 29 | C9 | C1 | C2 | C3 | −0.1 | −0.2 | 60 | C25 | C22 | C23 | C24 | −177 | −179.96 |
| 30 | C9 | C1 | C2 | C12 | −178.3 | −179.1 | 61 | C21 | C22 | C25 | O2 | 177 | 179.86 |
| 31 | C11 | C1 | C2 | C3 | 179.8 | 179.88 | 62 | C23 | C22 | C25 | O1 | 177 | 179.86 |

The XRD experimental data are in excellent agreement with the results of DFT calculations; as seen in Figure 3, the bond lengths in both DFT and XRD reflected a very good agreement (Figure 3a), with a 0.954 correlation coefficient (Figure 3b). Similarly, DFT/XRD angles reflected a higher degree of compatibility compared to the bond lengths (Figure 3c) with a 0.971 correlation coefficient (Figure 3d). DFT/XRD dihedral angles should reflect a

higher degree of compatibility compared to the angles (Figure 3e) with a 0.980 correlation coefficient (Figure 3f). Figure 3 reflected a high congruence between XRD and DFT analysis: ≈95% in the case of bonds, ≈97% in the case of angles, and ≈98% in the case of dihedral angles. The slight difference in the bonds, the angles, and dihedral angles can be attributed to the dynamic changes in bonds lengths and the angle values due to the difference in the phases. As in the DFT, the freedom to elongate the bonds and change the angle values is much greater in the gas state compared to the XRD solid state. Moreover, the absence of the internal molecular forces between the molecules in DFT may give both the angles and the bonds greater space and dynamic freedom compared to the XRD packed solid-state, which is with various intermolecular forces, resulting in a rigid lattice.

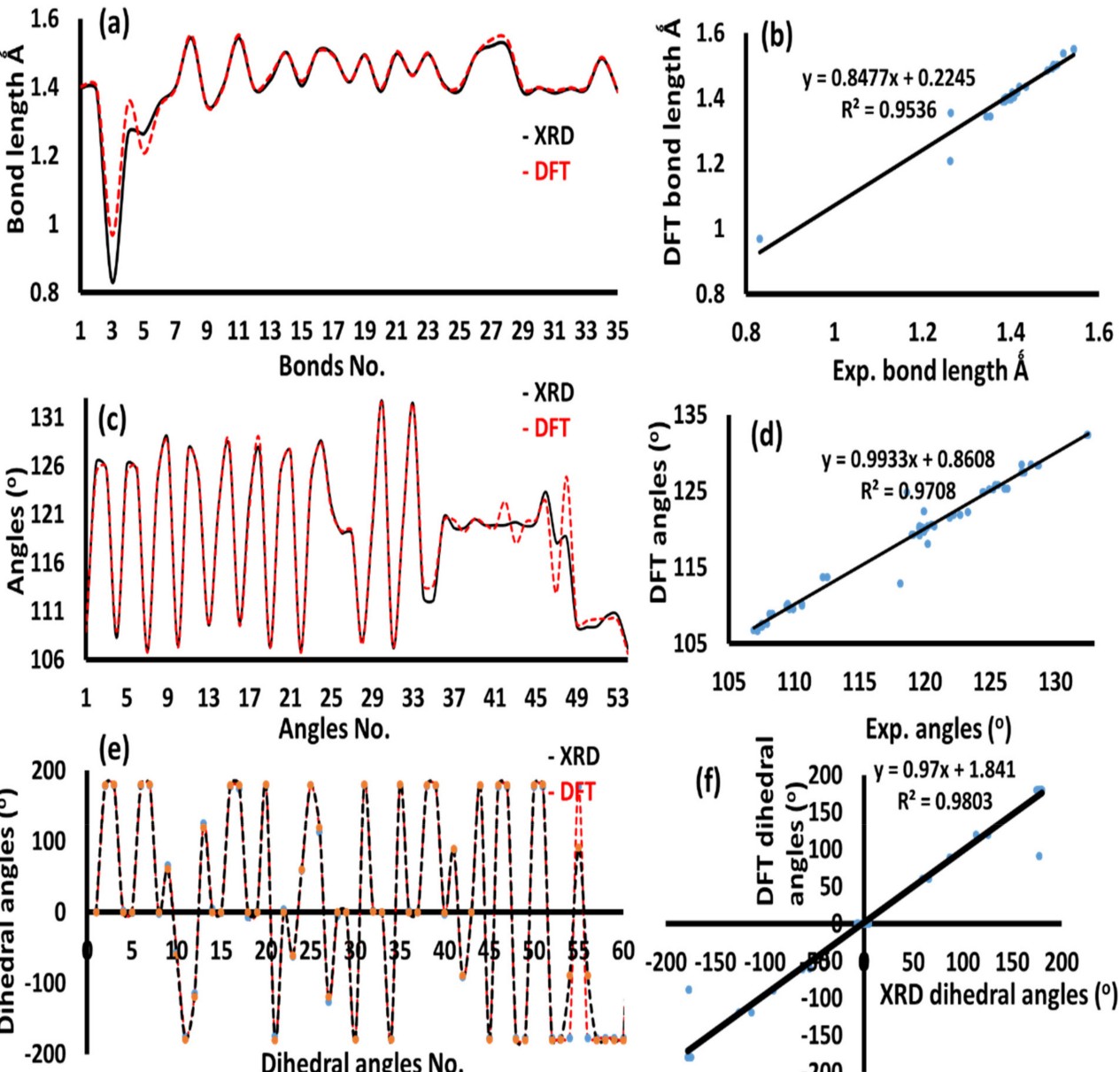

**Figure 3.** (**a**) Histogram of XRD/DFT bond lengths and its (**b**) correlation coefficient, (**c**) Histogram of XRD/DFT angles and its (**d**) correlation coefficient, (**e**) Histogram of XRD/DFT dihedral angles and its (**f**) correlation coefficient.

### 3.3. XRD Packing and HSA Investigation

In the packing mode of the desired B-complex crystal, a "tail-to-tail" dimer interaction was detected via X-ray and computed by HSA for the first time, as seen in Figure 4. For each molecule tail, two short hydrogen bonds of type O-H . . . .O=C with 1.788 and 1.870 Å formed a very stable 2D-S8 synthon dimer (Figure 4a). Two types of $C_{Me}$-H . . . ..F-B H-bonds with 2.658 Å formed 2D-S12 synthon, as seen in Figure 4b; there were another four $C_{ring}$H . . . .O=C H-bonds: two with 2.659 Å that formed 1D supramolecular extensions and two with 2.699 Å that formed 2D-S10 synthon, as seen in Figure 4c. No C-H...π or π–π stacking connections were detected in the complex lattice.

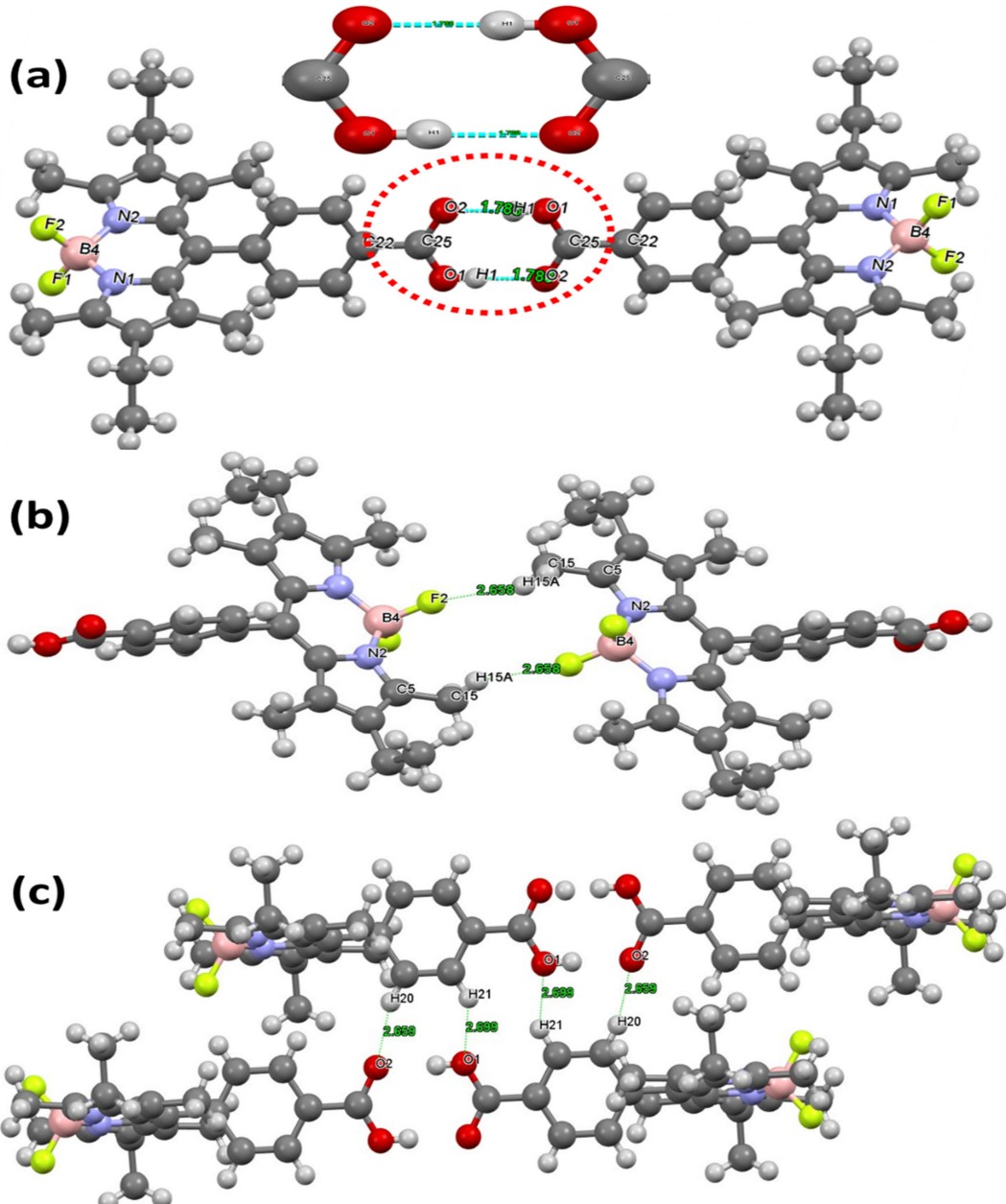

**Figure 4.** (**a**) O-H . . . .O, (**b**) $C_{Me}$-H . . . ..F, and (**c**) $C_{ring}$H . . . .O H-bonds interactions.

To understand the molecule mode surface interactions with the surrounding molecules [36–47], HSA computation was carried out in the 0.89 to 1.98 a.u. range, as shown in Figure 5. The $d_{norm}$ reflected the presence of two large red spots that are consistent with H . . . .O hydrogen bonds types only; the H . . . .F hydrogen bond detected by XRD

was not found by HSA (Figure 5a). No C-H...π or π–π stacking connections were detected in the shape index, as seen in Figure 5b. Accordingly, the results of HSA supported well the XRD-packing outcome. Furthermore, the inside $H_{atom}$ ... $All_{atom}$ outside two diminution-fingerprint plots contact ratios are illustrated in Figure 5c. The H ... H interactions were found to have the highest contributions (70.6%); meanwhile, the H ... .B was found to have the lowest contributions, and the other H ... X interactions were illustrated in [H ... C>H ... F>H ... O>H ... N] order.

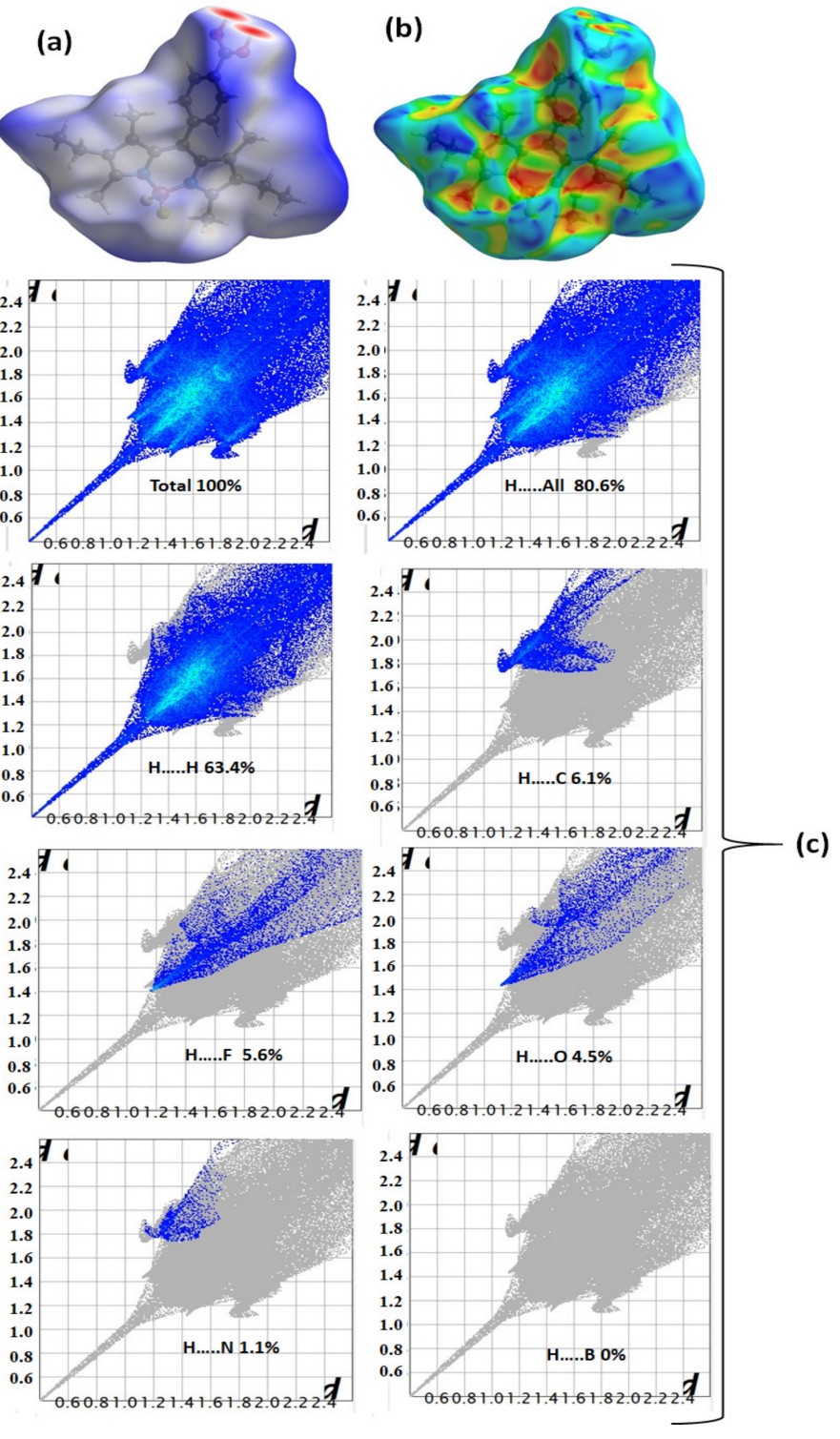

**Figure 5.** (**a**) $d_{norm}$ HSA map, (**b**) shape index, and (**c**) 2D-FP plots.

### 3.4. MEP, Charges, and GRD Investigations

The MEP, MAC, and NPA calculation reflected the presence of nucleophilic/electrophilic centers at the surface, as shown in Figure 6. For example, the MEP, showed the O and F atoms as nucleophilic centers with the red color; meanwhile, the blue color labeled the H of carboxylic acid as very strong electrophile sites, and a couple of Hs belong to Me and Ph as electrophilic centers; the positive and negative charges at H and O in the carboxylic part supported the tail-to-tail hydrogen bonds interactions, as seen in Figure 6a. NPA and MAC charge population charges showed O and F atoms with negative charge (Figure 6b and Table 3). Moreover, the B and all hydrogen atoms are with positive charge, and the H4, which is the H of the carboxylic group, possessed the highest positive charge 0.265e MAC and 0.468e for NPA (Table 3). A good correlation coefficient between NPA/MAC charges plotted with a 0.9041 value has been observed, as seen in Figure 6c. The MPE, NPA, and MAC results are in a high degree of correlation with the HSA computed as well as the XRD packing.

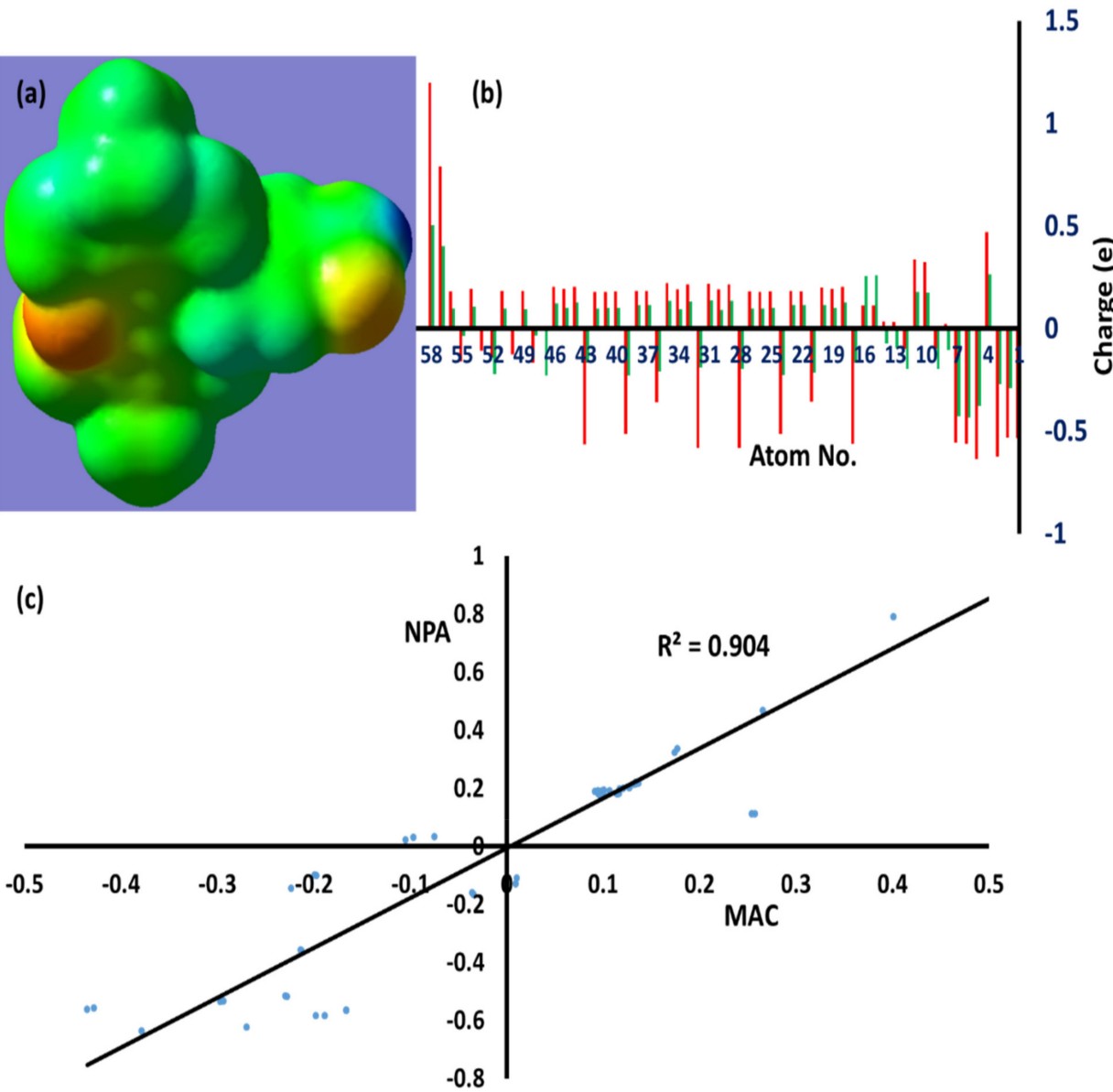

**Figure 6.** (**a**) MEP at B3LYP/6311G(d,P) level, (**b**) NPA (- - -) and MAC (- - -) charge, and (**c**) NPA/MAC graphical correlation.

**Table 3.** MAC and NPA charge population.

| No. | Atom | MAC | NPA | No. | Atom | MAC | NPA |
|-----|------|------|------|-----|------|------|------|
| 1 | F | −0.29709 | −0.53585 | 30 | H | 0.091309 | 0.18918 |
| 2 | F | −0.29348 | −0.53248 | 31 | H | 0.13651 | 0.2186 |
| 3 | O | −0.26969 | −0.62354 | 32 | C | −0.18864 | −0.58366 |
| 4 | H | 0.265325 | 0.46816 | 33 | H | 0.131435 | 0.2136 |
| 5 | O | −0.37872 | −0.63625 | 34 | H | 0.094352 | 0.19069 |
| 6 | N | −0.43505 | −0.56227 | 35 | H | 0.133131 | 0.22047 |
| 7 | N | −0.4279 | −0.55671 | 36 | C | −0.2127 | −0.3594 |
| 8 | C | −0.10506 | 0.02207 | 37 | H | 0.113849 | 0.18236 |
| 9 | C | −0.19772 | −0.09985 | 38 | H | 0.116174 | 0.18306 |
| 10 | C | 0.17403 | 0.32414 | 39 | C | −0.2293 | −0.51552 |
| 11 | C | 0.176512 | 0.33589 | 40 | H | 0.098476 | 0.1797 |
| 12 | C | −0.19924 | −0.09794 | 41 | H | 0.099777 | 0.17848 |
| 13 | C | −0.09659 | 0.03016 | 42 | H | 0.097746 | 0.17814 |
| 14 | C | −0.07529 | 0.03442 | 43 | C | −0.16669 | −0.56563 |
| 15 | C | 0.257276 | 0.11308 | 44 | H | 0.126768 | 0.20287 |
| 16 | C | 0.253713 | 0.11242 | 45 | H | 0.100512 | 0.19281 |
| 17 | C | −0.16645 | −0.56321 | 46 | H | 0.120702 | 0.20111 |
| 18 | H | 0.127112 | 0.20256 | 47 | C | −0.23101 | −0.00949 |
| 19 | H | 0.098994 | 0.19184 | 48 | C | −0.03429 | −0.16719 |
| 20 | H | 0.116622 | 0.19892 | 49 | H | 0.094639 | 0.18222 |
| 21 | C | −0.21392 | −0.35705 | 50 | C | 0.009118 | −0.12842 |
| 22 | H | 0.11481 | 0.18151 | 51 | H | 0.095902 | 0.18206 |
| 23 | H | 0.114148 | 0.18229 | 52 | C | −0.22343 | −0.14409 |
| 24 | C | −0.22818 | −0.51663 | 53 | C | 0.010034 | −0.10858 |
| 25 | H | 0.100367 | 0.17925 | 54 | H | 0.106701 | 0.19256 |
| 26 | H | 0.095844 | 0.1774 | 55 | C | −0.03609 | −0.16044 |
| 27 | H | 0.097458 | 0.17899 | 56 | H | 0.097616 | 0.18146 |
| 28 | C | −0.19803 | −0.58381 | 57 | C | 0.400776 | 0.79126 |
| 29 | H | 0.133892 | 0.21604 | 58 | B | 0.502918 | 1.19828 |

The GRD quantum parameters such as chemical potential ($\mu$), the Electrophilicity ($\omega$), Hardness ($\eta$), Softness ($\sigma$), and Electronegativity ($\chi$) of the B-complex were calculated using the equations listed in Table 4.

| | |
|---|---|
| *I: Ionization potential* $= -E_{HOMO}$ | (1) |
| A: *Electron affinity* $= -E_{LUMO}$ | (2) |
| $\Delta E_{gap}$: *Energy gap* $= E_{HOMO} - E_{LUMO}$ | (3) |
| $\chi$: *Absolute electronegativity* $= (I + A)/2$ | (4) |
| $\eta$: *Global hardness* $= (I - A)/2$ | (5) |
| $\sigma$: *Global softness* $= 1/\eta$ | (6) |
| $\mu$: *Chemical potential* $= -\chi$ | (7) |
| $\omega$: *Electrophilicity* $= \mu^2/2\eta$ | (8) |

**Table 4.** Calculated GRD quantum parameters.

| GRD | | Value |
|-----|---|-------|
| Global total energy | $E_T$ | −1415.9090 a.u, |
| Low unoccupied molecular orbital | LUMO | −0.0838 a.u. |
| High occupied molecular orbital | HOMO | −0.1917 a.u. |
| Energy gap | $\Delta E_{gap}$ | 0.1080 a.u. 2.941 eV |
| Electron affinity | A | 2.2803 eV |
| Ionization potential | I | 5.2164 eV |
| Global hardness | $\eta$ | 2.9361 eV |
| Global softness | $\sigma$ | 0.3406 eV |
| Chemical potential | $\mu$ | −3.7484 eV |
| Absolute electronegativity | X | 3.7484 eV |
| Electrophilicity | $\omega$ | 2.3927 eV |
| Dipole Moment | $\mu$ | 3.0497 D |

### 3.5. FT-IR and DFT-IR Spectroscopy

The experimental FT-IR and DFT-IR calculations spectra of the [N∩NBF$_2$] complex are illustrated in Figure 7a,b respectively.

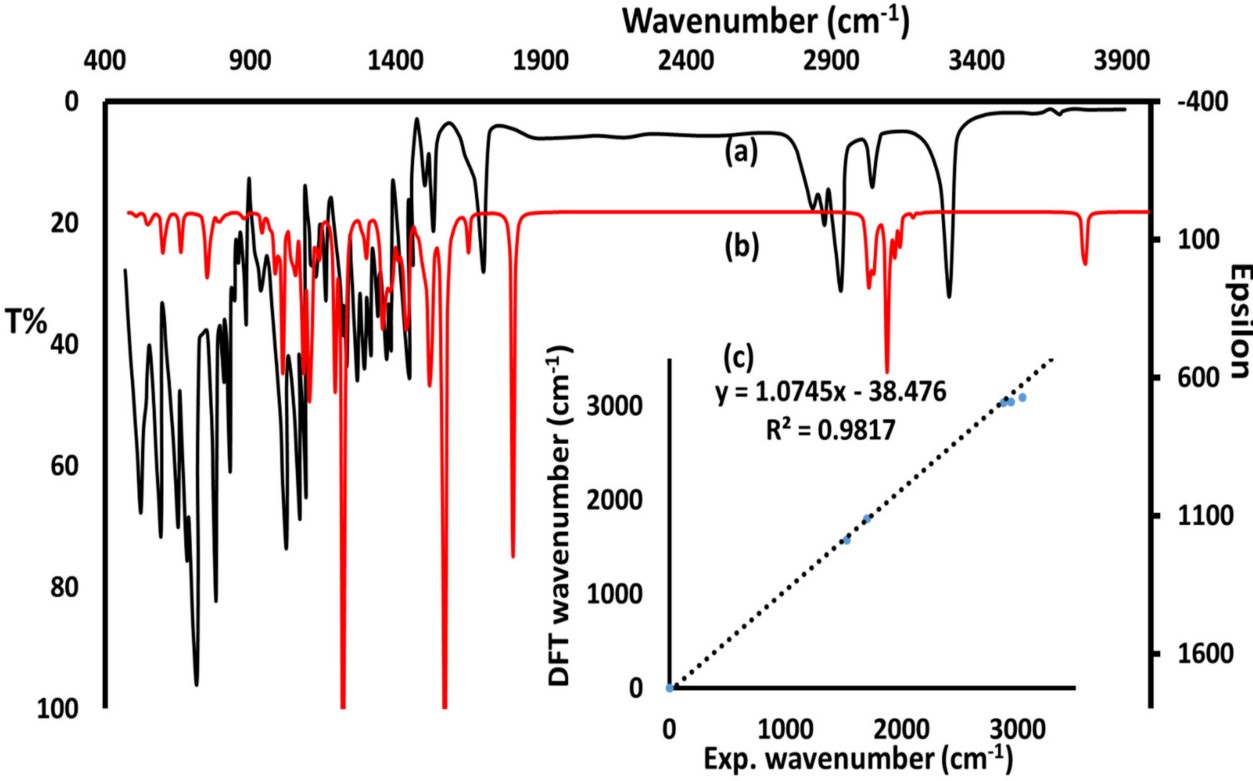

**Figure 7.** (**a**) Exp. FT-IR, (**b**) DFT-IR, and (**c**) Exp./DFT-IR correlation.

In general, several functional groups vibrations that recognized the structure of the desired complex have been recorded. The main functional groups and their exp. and thero. wavenumber values are illustrated as $v$COOH (exp. at 3310 cm$^{-1}$ and DFT at 3770 cm$^{-1}$), $v$C-H$_{ph}$ (exp. at 3040 cm$^{-1}$ and DFT at 3090–3140 cm$^{-1}$), $v$C-H$_{alhyl}$ (exp. at 2860–2950 cm$^{-1}$ and DFT at 3000–3040 cm$^{-1}$), $v$C=O (exp. at 1705 cm$^{-1}$ and DFT at 1800 cm$^{-1}$), $v$C=N (exp. at 1550 cm$^{-1}$ and DFT at 1570 cm$^{-1}$), $v$B-N (exp. at1290 cm$^{-1}$ and DFT at 1355 cm$^{-1}$), and $v$B-F (exp. at 1095 cm$^{-1}$ and DFT at 1220 cm$^{-1}$). A high degree of compatibility demonstrated by the plotting together of experimental and DFT wavenumbers with 0.972 graphical correlation was recorded as seen in Figure 7c.

### 3.6. DOS, HOMO→LUMO, e-transfer/TD-SCF/DFT/B3LYP, and Solvents Effect

Figure 8 shows the HOMO and LUMO shapes, energy, and DOS analysis, which were theoretically stimulated in MeOH. The calculations support electron transfer from HOMO→LUMO with $\Delta E_{HOMO/LUMO}$ = 2.94 eV (Figure 8a), the $\Delta E$ value was also calculated via DOS as another method and found to be 2.95 eV (Figure 8b). Both $\Delta E_{HOMO/LUMO}$ and $\Delta E_{DOS}$ energy values correspond to an electronic transition in the visible region (≈430 nm). The absorption behavior of the desired B-complex was recorded by UV-visible spectroscopy using an MeOH solvent (Figure 8c). The electron transfers in solution revealed four bands with $\lambda_{max}$ 255, 308, 363, and 430 nm values, which are assigned to $\pi \rightarrow \pi^*$ e-transition localized in the polyheteroaromatic skeleton of the N∩N-ligand. The $\pi - \pi^*$ transition bands in the complex are in agreement with TD-DFT computations (Figure 8c and Table 5). By applying the TD-DFT at RT and using same solvent, four main bands with $\lambda_{max}$ 255, 295, 367, and 433 nm values are predicted Figure 8c. The DFT calculations predicted that the lowest energy transition exhibits the highest oscillator strength and

corresponds to the electronic transition from HOMO to the LUMO with 433.5 nm, which compares to the experimentally determined absorption at 430 nm. The solvent effect analysis on this absorption band exhibited a blue shift (22 nm) upon increasing the polarity. For a deeper understanding of the electron transfer in the [N∩NBF$_2$] complex, CAM-TD-DFT calculations were carried out using methanol and the same level of calculations as shown in Figure 8c and Table 6. CAM-TD-DFT reflected also four main bands with $\lambda_{max}$ 225, 254, 325, and 410 nm values; in general, good agreement between the CAM-TD-DFT and TD-DFT was recorded, noting that CAM-TD-DFT showed less wavelength shifts values of their bands and TD-DFT wavelength values are closer to the experimental UV-vis result, as can be seen from Figure 8.

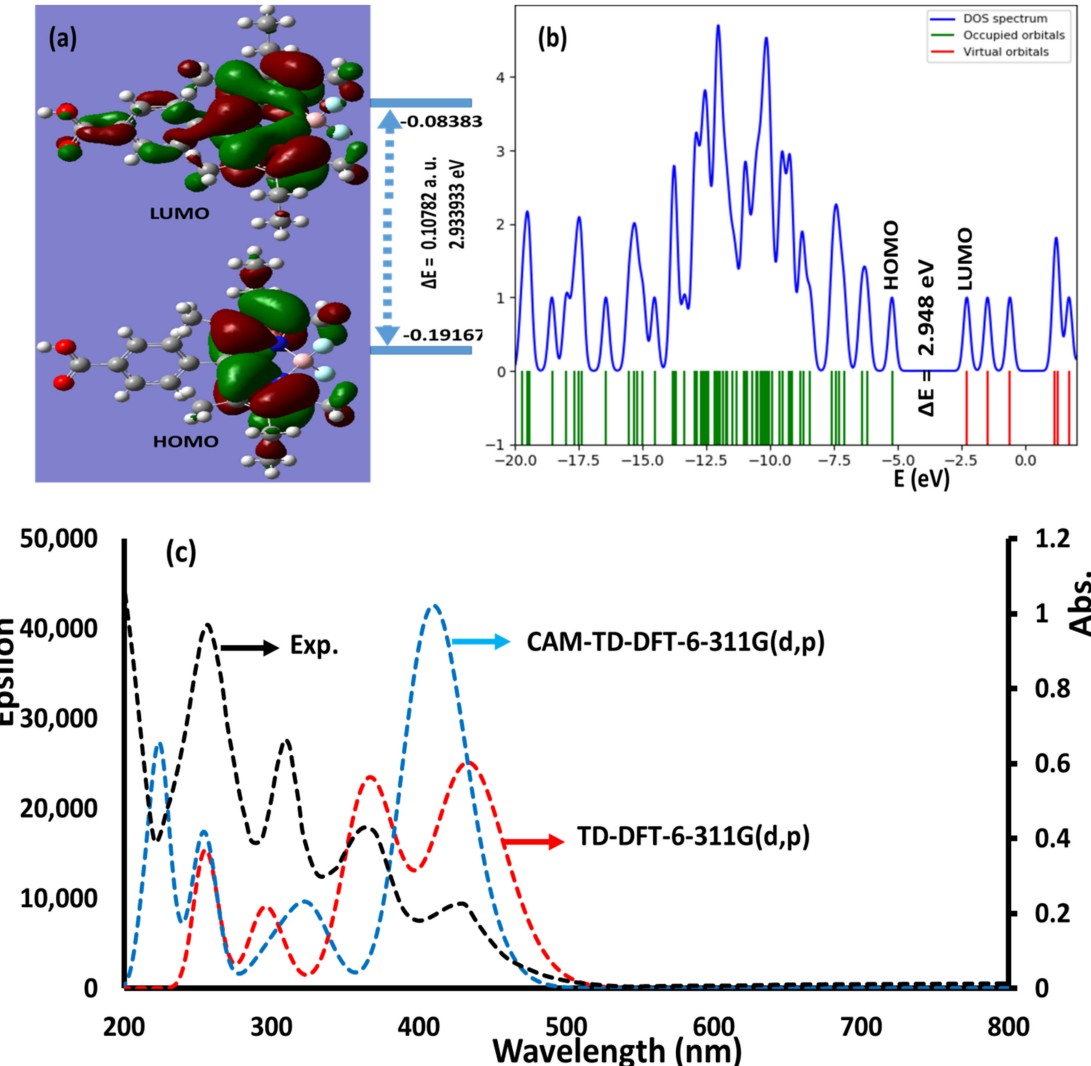

**Figure 8.** (**a**) HOMO/LUMO, (**b**) DOS, and (**c**) experimental, TD-DFT, and CAM-TD-DFT spectra in MeOH.

The experimental and TD-DFT solvents' effect on the electron transition in [N∩NBF$_2$] was evaluated as seen in Figure 9. Due to the poor solubility of the complex, the study was limited to DMSO, CH$_3$CN, CHCl$_3$, and MeOH solvents. Experimentally, no changes on the wavelengths but only a slight change in the intensity of the four packs were detected by changing the solvents, as can be seen in Figure 9. In the TD-DFT, only the two internal peaks have slight changes on the wavelengths and the intensity of the band; the terminal peaks have no effect by changing the solvents, as seen in Figure 9.

**Table 5.** TD-DFT computations parameters.

| No. | $\lambda_{max}$ (nm) | Osc. Str. (f) | Major Contributions |
|---|---|---|---|
| 1 | 433.49 | 0.3433 | HOMO->LUMO (84%) |
| 2 | 389.14 | 0.0127 | H-1->LUMO (34%), HOMO->L+1 (65%) |
| 3 | 367.63 | 0.2794 | H-1->LUMO (56%), HOMO->LUMO (16%), HOMO->L+1 (26%) |
| 4 | 354.34 | 0.0495 | H-2->LUMO (97%) |
| 5 | 308.29 | 0.0034 | HOMO->L+2 (97%) |
| 6 | 296.94 | 0.009 | H-1->L+1 (98%) |
| 7 | 295.84 | 0.113 | H-3->LUMO (88%) |
| 8 | 287.27 | 0.0002 | H-4->LUMO (45%), H-4->L+1 (51%) |
| 9 | 282.25 | 0.0012 | H-2->L+1 (99%) |
| 10 | 277.55 | 0.0027 | H-5->LUMO (97%) |
| 11 | 259.69 | 0.0016 | H-4->LUMO (54%), H-4->L+1 (42%) |
| 12 | 255.00 | 0.2109 | H-6->LUMO (73%), H-3->L+1 (14%) |

**Table 6.** CAM-TD-DFT computations parameters.

| No. | $\lambda_{max}$ (nm) | Osc. Str. (f) | Major Contributions |
|---|---|---|---|
| 1 | 409.5942947 | 0.5877 | HOMO->LUMO (95%) |
| 2 | 326.47 | 0.1111 | H-1->LUMO (93%) |
| 3 | 306.38 | 0.0594 | H-2->LUMO (95%) |
| 4 | 290.05 | 0.0263 | HOMO->L+1 (95%) |
| 5 | 258.94 | 0.0032 | H-6->L+1 (76%), H-6->L+4 (10%) |
| 6 | 254.11 | 0.2336 | H-3->LUMO (93%) |
| 7 | 240.04 | 0.0105 | H-4->LUMO (51%), H-4->L+1 (29%), H-3->L+2 (15%) |
| 8 | 237.18 | 0.0224 | HOMO->L+2 (98%) |
| 9 | 224.29 | 0.3417 | H-5->LUMO (77%), H-3->L+1 (12%) |
| 10 | 221.17 | 0.0029 | H-4->LUMO (39%), H-4->L+1 (28%), H-1->L+1 (23%) |
| 11 | 218.89 | 0.0033 | H-4->L+1 (12%), H-1->L+1 (72%) |
| 12 | 213.61 | 0.116 | H-5->LUMO (13%), H-3->L+1 (64%) |

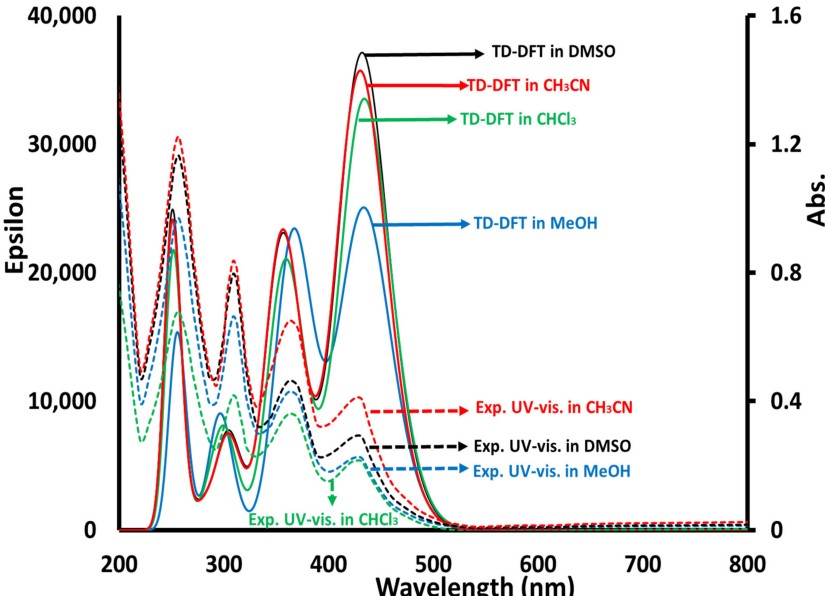

**Figure 9.** Experimental and TD-DFT solvents effect.

## 4. Conclusions

The [N∩NBF$_2$] difluoride 10-(4-carboxyphenyl)-2,8-diethyl-5,5-difluoro-1,3,7,9-tetra methyl-5H-dipyrrolo[1,2-c:2′,1′-f][1,3,2]diazaborinin-4-ium-5-uide B-complex was synthesized in very good yield. The NMR, IR, DFT, and XRD data proved the formation of tetrahedral geometry around the Boron center in the targeted complex. The $^1$H, $^{11}$B, and $^{19}$F-NMR successfully supported the symmetrical $\pi$-extended phenomena in the BODIPY complex. The XRD/HSA interactions reflected the presence of 2H . . . .O tail-to-tail carboxylic dimer as well as H . . . .F and non-classical H . . . .O H-bonds interactions in the lattice of the B-complex. The DFT/B3LYP/6-311G(d,p) angles and bond distances' structural parameters were found to be very consistent with XRD parameters. MAC, NPA, GRD, and MEP reflected the presence of both nucleophilic and electrophilic centers in the B-complex. With the help of DOS, HOMO/LUMO, Exp./IR-DFT, and TD-DFT/UV-vis. behaviors of the tetrahedral B-complex were well elucidated.

**Author Contributions:** Formal analysis, A.A.A. (Abdulrahman A. Alsimaree), N.I.A. and A.A.A. (Abeer A. AlObaid); data curation, O.M.A.; supervision, J.G.K.; validation, M.M.; review and editing, A.Z.; writing I.W. All authors have read and agreed to the published version of the manuscript.

**Funding:** Not applicable.

**Data Availability Statement:** Not applicable.

**Acknowledgments:** The authors extend their appreciation to Paul Gordon Waddell of Newcastle University for his collaboration in the collection of the single crystal X-ray data of the complex.

**Conflicts of Interest:** The authors declare no conflict of interest.

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
