# Peer review of "π-Extended Boron Difluoride [NՈNBF2] Complex, Crystal Structure, Liquid NMR, Spectral, XRD/HSA Interactions: A DFT and TD-DFT Study"

_crystals, doi:10.3390/cryst11060606_

Round 1

Reviewer 1 Report

In the submitted study the Authors have obtained a new crystal structure and done some spectroscopic analysis  and theoretical calculations on a single complex. The calculations are routine and should be improved, i.e. by including periodicity and intermolecular interactions in the solid state. Therefore, I recommend major revision with my suggestions presented below.

In the introduction the Authors write a lot about the previously reported similar complexes. However, the one important thing is not stated clearly-was the compound, that is the object of the current study, obtained before? I am not asking about its crystal structured but I would like to know if this particular complex has been synthesized before. If no, it should be stated clearly. If yes it should be described in the introduction and the previously obtained results should be compared with ones presented in this study.

Computational. The Authors have submitted their manuscript to Crystals. What is more, they have solved the crystal structure and therefore had the access to the cif file. My question is, why the Authors have nor performed periodic DFT calculations (in the solid state) using many of the available software, i.e. CASTEP, CRYSTAL, etc. Instead, the Authors have performed gas-state calculations and compare the results with the experimental solid state (crystal). This can be the cause of the differences between the experimental and computational results.

Line 64, basis set, what about the polarization functions on heavy atoms and hydrogen?

Line 71, it should be described in details which spectrometer has been used for which NMR analysis. Further, more experimental details on NMR analysis are required, i.e. pulse sequence, number of scans etc.

Table 1, what was the Z’?

Figure 1, the axes are not described correctly (i.e. “chemical shift, [ppm]).

The Authors have performed DFT calculations and NMR analysis. Why haven’t you calculated NMR shieldings in order to 1)confirm the spectral assignment 2)confirm the optimized geometry?

Figure 3, what about dihedrals? The Authors should compare them too.

Figure 3b and 3d, there are some major differences, I think at least some of them need to be explained. Think about the dynamics.

Figure 3b and 3d, the function formula should be provided in the form of y=ax+b.

Figure 7c, the Authors are comparing the experimental (in MeOH) and theoretical (in gas phase) spectra. Why the DFT calculations haven’t been done using PCM?

Author Response

Reviewer 1

1- In the introduction, the Authors write a lot about the previously reported similar complexes. However, the one important thing is not stated clearly-was the compound, that is the object of the current study, obtained before? I am not asking about its crystal structured but I would like to know if this particular complex has been synthesized before. If no, it should be stated clearly. If yes it should be described in the introduction and the previously obtained results should be compared with ones presented in this study.

The complexes were synthesized recently as we mention in reference 28, but we here report the XRD-crystal/DFT in addition to confirming the π‑extended boron difluoride phenomena in [NՈNBF2] complex by NMR. We made this point very clear by adding the following sentence to the introduction.  Recently, the same complex and its analogs were prepared by Shipalova et al, the author's studies focused in the pH and polarity fluorescent molecular sensorics involving BODIPY ligand. Moreover, none of the complex structures were solved by XRD-crystal nor the π‑extended boron difluoride phenomena were studied [28].

2- Computational. The Authors have submitted their manuscript to Crystals. What is more, they have solved the crystal structure and therefore had the access to the cif file. My question is, why the Authors have nor performed periodic DFT calculations (in the solid state) using many of the available software, i.e. CASTEP, CRYSTAL, etc. Instead, the Authors have performed gas-state calculations and compare the results with the experimental solid state (crystal). This can be the cause of the differences between the experimental and computational results.

We are doing our calculation using Gaussian09 program that is one of the best and trustable program, to change the program is a good suggestion but we do not have access or knowledge to CASTEP, CRYSTAL, or even to other programs, it will be a good point in the future work to compare the Gaussian09 result with other programs.

3-Line 64, basis set, what about the polarization functions on heavy atoms and hydrogen?

We are working with B complexes and as you know no heave metal centers were used, therefore, DFT/B3LYP/6-311G(d,p) found to be suitable for B-complex since B is a light element.

4-Line 71, it should be described in details which spectrometer has been used for which NMR analysis. Further, more experimental details on NMR analysis are required, i.e. pulse sequence, number of scans etc.

The NMR use performed on JEOL ECS 400 Bruker Advance 300, Jeol Lambda 500, or a Bruker Advance 700 MHz instrument using CDCl3 as solvent at RT. Since the elements H, B, and F which NMRed her are with high natural abundance <99% therefore classical NMR was performed with 90° pulse, and 8 to 32 number of scans were used, TMS reference for 1H-NMR, CFCl3 reference for 19F-NMR and BF3.OEt2 for 11B-NMR.

5-Table 1, what was the Z’?

Z′ is defined as the number of formula units in the crystallographic unit cell divided by the number of independent general positions.

6-Figure 1, the axes are not described correctly (i.e. “chemical shift, [ppm]).

Corrected accordingly.

7-The Authors have performed DFT calculations and NMR analysis. Why haven’t you calculated NMR shieldings in order to 1)confirm the spectral assignment 2)confirm the optimized geometry?

GiAO-DFT/B3LYP/6-311G(d,p) NMR was performed using the same references and solvent, the result was inserted to Fig. 1 together with their experimental relatives. The difference in the theoretical calculations is not surprising; because in 1H-NMR (Fig. 1b) the DFT NMR cannot feel the π-extended phenomena, therefore, the BODIPY in [NՈNBF2] complex become asymmetry ligand and have no-C2-plane of symmetry that increased the protons type. Meanwhile in 11B-NMR (Fig. 1d.) and  19F (Fig. 1f.) the DFT neglected the effect of the two nuclei on each other, therefore no splitting was detected resulting a simple system, singlet for B and doublet for F atoms.

I personally love to remove the GiAO-DFT/B3LYP/6-311G(d,p) NMR result.

8- Figure 3, what about dihedrals? The Authors should compare them too.

Corrected accordingly, please see Table 2  and Fig.3e and 3f

9-Figure 3b and 3d, there are some major differences, I think at least some of them need to be explained. Think about the dynamics.

Fig. 3 reflected a high congruence between XRD and DFT analysis ~ 95% in the case of bonds, ~ 97% in the case of angles, and ~ 98% in the case of dihedral angles. The slight difference in the bonds, the angles, and dihedral angles can be attributed to the dynamic changes in bond lengths and the angle values due to the difference in the phases. As in the DFT, the freedom to elongate the bonds and change the angle values is much greater in the gas state compared to the XRD-solid state. Moreover, the absence of the internal molecular forces between the molecules in DFT may give both the angles and the bonds greater space and dynamic freedom compared to the XRD-packed solid-state which is with various intermolecular forces resulting a rigid lattice.

10- Figure 3b and 3d, the function formula should be provided in the form of y=ax+b.

Corrected accordingly.

11-Figure 7c, the Authors are comparing the experimental (in MeOH) and theoretical (in the gas phase) spectra. Why the DFT calculations haven’t been done using PCM?

This is a good suggestion, but usually, the simpler theory like TD-DFT is applied in the event that there is an acceptable degree of agreement, as in our case; there is no need to study a more complex and hybrid theory like PCM or CAM TD-DFT.

Reviewer 2 Report

The article entitled "π‑extended boron difluoride [NՈNBF2] complex, crystal structure, liquid NMR, spectral, XRD/HSA-interactions: A DFT and TD-DFT study" deals with the synthesis of new BODIPY complexes and their characterization by NMR, XRD UV vis and also theorical study by DFT.

The article is well written but it remains few problem before publications.

The authors describes the BODIPY as new, but they refer to an publications when they deals with synthesis -> it is not a new compounds.

The authors show H-NMR, B-NMR and F-NMR, the authors can also put C-NMR to complete the study.

Moreover they don't put the integration for the H-NMR, helping the comprehension about the symmetry of the molecules.

With these modifications, the article can be published.

Author Response

Thank you for your time and for accepting the review of our MS.

The article entitled "π‑extended boron difluoride [NՈNBF2] complex, crystal structure, liquid NMR, spectral, XRD/HSA-interactions: A DFT and TD-DFT study" deals with the synthesis of new BODIPY complexes and their characterization by NMR, XRD UV vis and also theorical study by DFT.

The article is well written but it remains few problem before publications.

  • The authors describes the BODIPY as new, but they refer to an publications when they deals with synthesis -> it is not a new compounds.

Recently, the same complex and its analogs were prepared by Shipalova et al, the author's studies focused in the pH and polarity fluorescent molecular sensorics involving BODIPY ligand. Moreover, none of the complex structures were solved by XRD-crystal nor the π‑extended boron difluoride phenomena were studied [28].

  • The authors show H-NMR, B-NMR, and F-NMR, the authors can also put C-NMR to complete the study.

This is correct sir, but unfortunately; due to the weak solubility of such complexes in deuterated solvents, unfortunately, we are not able to report a good C-NMR spectrum.

  • Moreover, they don't put the integration for the H-NMR, helping the comprehension about the symmetry of the molecules.

The integrated H-NMR, original figure was inserted now to the MS see Fig. 1a.

Round 2

Reviewer 1 Report

Unfortunately, I still can’t recommend the publication of this manuscript in Crystals, I will justify my decision below.

  1. The Authors have made its clear now that the compound has been obtained before. This is fine, however if the synthesis is not new than the stress should be put on the other parts of the study.
  2. Computations, I agree that Gaussian is reliable and most popular software used for molecular modelling of NON PERIODIC structures. Nowadays, the calculations for periodic crystal structures should be done using periodic DFT software listed in my previous review. In my previous report when I wrote about heave atoms I was thinking about atoms other than H, as usually referred in the computational chemistry.
  3. The Authors write “to change the program is a good suggestion but we do not have access or knowledge to CASTEP, CRYSTAL, “ this is not an excuse.
  4. The solution NMR is not described properly. You can not write “the NMR analysis has been done on Bruker 300 OR Jeol OR Bruker 700”-do I have to choose or guess? The description of the methodology should be clear and detailed.
  5. I know what a “Z ‘ “ is but I have requested for that information to be added to Table 1.
  6. The Authors write “ there is no need to study a more complex and hybrid theory like PCM “ The PCM is not a theory, it is a way (one of many but most popular) to include the solvent effect in the DFT calculations and this is mandatory when the Authors want to compare the results of calculations with the experimental results in the liquid state.
  7. Finally, the Authors study ONE (1) small compound that is not new and has been obtained before. The only new part in this study is its crystal structure and routine NMR analysis. The calculations are not very informative due to the reasons listed above. So my recommendations are as follow: either increase the level of calculations (periodic DFT) and add some more experimental data (i.e. ssNMR, FT-IR, Raman) OR increase the number of compounds OR look for a lower quality journal.
  8. The quality of Figure 1 is very low.

Author Response

We thank you Sir for your keenness to get the work out in better quality.

  1. The Authors have made it clear now that the compound has been obtained before. This is fine, however if the synthesis is not new than the stress should be put on the other parts of the study.

Thank you.

  1. Computations, I agree that Gaussian is reliable and most popular software used for molecular modelling of NON PERIODIC structures. Nowadays, the calculations for periodic crystal structures should be done using periodic DFT software listed in my previous review. In my previous report when I wrote about heave atoms I was thinking about atoms other than H, as usually referred in the computational chemistry.

Thanks sir all the atom are H, B, C , N and O all are light elements and DFT/B3LYP/6-311G(d,p) is suitable.

  1. The Authors write “to change the program is a good suggestion but we do not have access or knowledge to CASTEP, CRYSTAL, “ this is not an excuse.

I agree it is no excuse but we like the others used to use Gussian09 which is good enough and the most popular and trustable program.

The solution NMR is not described properly. You can not write “the NMR analysis has been done on Bruker 300 OR Jeol OR Bruker 700”-do I have to choose or guess? The description of the methodology should be clear and detailed.

Sorry, I forget to delete the second instrument description,

The NMR details were illustrated as

The NMR was performed on JEOL ECS 400 MHz Bruker Advance 300 instrument using CDCl3 as solvent at RT. 10 mg of the complex powder was suspended in 3 ml of CDCl3, the clean solution was decanted to 3mm Wilmad NMR tube and filled up to 3 cm length to be used for the H, B, and F NMR. Since the element H, B, and F which NMRed are with high natural abundance <99% therefore classical NMR was performed with 90° pulse and 8 to 32 number of scan and TMS reference for 1H-NMR, CFCl3 reference for 19F-NMR and BF3.OEt2 for 11B-NMR was used.  

I know what a “Z ‘ “ is but I have requested for that information to be added to Table 1.

Was inserted to Table 1, Sorry for the wrong understanding.

The Authors write “ there is no need to study a more complex and hybrid theory like PCM “ The PCM is not a theory, it is a way (one of many but most popular) to include the solvent effect in the DFT calculations and this is mandatory when the Authors want to compare the results of calculations with the experimental results in the liquid state.

We have added CAM-TD-DFT in addition to TD_DFT and compared to the exp. Uv-vis. new figure and table were added to the TXT. Moreover, Exp. and theoretical solvents effect using new four solvents were inserted to the TXT, new figure was also inserted.

Finally, the Authors study ONE (1) small compound that is not new and has been obtained before. The only new part in this study is its crystal structure and routine NMR analysis. The calculations are not very informative due to the reasons listed above. So my recommendations are as follow: either increase the level of calculations (periodic DFT) and add some more experimental data (i.e. ssNMR, FT-IR, Raman) OR increase the number of compounds OR look for a lower quality journal.

  • NMRDB and GIAO-DFT/B3LYP/6-311G(d,p) NMR were performed and compared to the experimental result using the same references and CDCl3 solvent see Fig. 1
  • Experimental/DFT-IR was also carried out and compared see the new figure 7.
  • Unfortunately, we don’t have a Raman instrument in our lab to compare its result with the theoretical one.
  1. The quality of Figure 1 is very low.

The quality enhanced up to our limit, hope it is now ok otherwise no problem to delete it.

once more thank you